# Low-Protein Infant Formula and Obesity Risk

**DOI:** 10.3390/nu14132728

**Published:** 2022-06-30

**Authors:** Stefanie M. P. Kouwenhoven, Jacqueline Muts, Martijn J. J. Finken, Johannes B. van Goudoever

**Affiliations:** 1Emma Children’s Hospital, Amsterdam UMC, Vrije Universiteit, University of Amsterdam, 1081 HV Amsterdam, The Netherlands; s.kouwenhoven@amsterdamumc.nl (S.M.P.K.); j.muts@amsterdamumc.nl (J.M.); 2Department of Dietetics, Erasmus MC, Sophia Children’s Hospital, Erasmus University, 3015 CN Rotterdam, The Netherlands; 3Department of Neonatology, Erasmus MC, Sophia Children’s Hospital, Erasmus University, 3015 CN Rotterdam, The Netherlands; 4Department of Pediatric Endocrinology, Emma Children’s Hospital, Amsterdam UMC, Vrije Universiteit, University of Amsterdam, 1081 HV Amsterdam, The Netherlands; m.finken@amsterdamumc.nl

**Keywords:** protein intake, early nutrition, infant nutrition, childhood obesity, amino acids

## Abstract

Infant formulas have been designed to mimic human milk for infants who cannot be breastfed. The overall goal is to establish similar functional outcomes to assure optimal growth, development, maturation of the immune system, and programming of the metabolic system. However, after decades of improving infant formula, growth patterns and body composition development are still different in formula-fed infants compared to breastfed infants, which could contribute to an increased risk of obesity among formula-fed infants. It has been hypothesized that the lower protein concentration of breast milk compared to infant formula influences infants’ growth and body composition. Thus, several trials in formula-fed infants with different protein intake levels have been performed to test this hypothesis. In this review, we discuss the current evidence on low-protein infant formula and obesity risk, including future perspectives and implications.

## 1. Background

Obesity in children and adolescents is associated with increases in blood pressure and blood glucose levels [1], rates of premature death [2], and the risk of coronary heart disease in adulthood [3]. Moreover, childhood obesity often tracks into adulthood [4], which, on its own, increases the risk of morbidity and mortality [5]. Therefore, childhood overweight and obesity can be considered major public health concerns.

Infant feeding may play a pivotal role in the risk of obesity in later life. The strongest and most consistent association for a protective, long-term effect is documented for breastfeeding, which is recommended for the prevention of obesity, cardiovascular diseases, and hypertension, as well as for the reduction of serum cholesterol in adulthood.

## 2. Could Lower Protein Intake Be a Mediator in the Reduction of Obesity?

The mechanism behind the protective effect of breastfeeding is not fully understood. Apart from potential demographic differences, several mechanisms may account for the lower obesity risk in breastfed infants compared to formula-fed infants, such as differences in appetite regulation, early growth patterns, circulating leptin, and the gut microbiome. Another candidate mechanism is early-life protein intake, which will be the focus of this article. The protein concentration in breast milk decreases over the weeks of lactation, while the protein concentration of infant formulas remains constant (Figure 1). It has been shown that protein intake during the first 6 months of life is up to 66–70% higher in formula-fed infants compared to breastfed infants [6]. The lower protein concentration of breast milk is hypothesized to be a factor that can influence infants’ growth, and lower protein intake could possibly prevent childhood obesity.

During the last decade, systematic reviews have been conducted to assess the long-term health effects of different protein intake levels [7,8,9]. These reviews show conflicting results regarding the beneficial effects of reduced protein intake early in life on obesity risk. The general conclusion is we need more data to provide a conclusive answer.

Accordingly, we performed a multicenter double-blinded trial, the ProtEUs study, in which the effects of a modified low-protein infant formula were assessed for both short-term [10] and long-term [11] growth and body composition. The ProtEUs study showed that the use of this infant formula with a reduced protein content of 20% was safe and supports adequate growth and body composition up until the age of 6 months [10]. Up until the age of 2 years, no differences in outcomes were found between the intervention and the control group, except for a temporal lower fat-free mass index at 4 months of age [11].

In addition and in contrast to the original hypothesis, a reduced protein intake during the first months of life did not affect glucose homeostasis or the insulin-like growth factor (IGF) system in formula-fed infants at the age of 4 months [12]. Since the anabolic hormones insulin and insulin-like growth factor-1 (IGF-1) are responsive to fluctuations in protein intake [13] and have growth-promoting properties [14,15], we expected to find lower insulin and IGF-1 levels in the modified low-protein group. However, there were no significant differences in both hormones between the two formula groups. In line with these results, no association between the IGF-1 level at 4 months and anthropometric outcomes until the age of 2 years was found.

So, our conclusion was that up until the age of 2 years, lowering protein intake during the first 6 months of life to values of around 9.2 g/day, when compared to 11.4 g/day, did not affect body composition. However, a lower protein intake during infancy could still be beneficial.

For instance, renal function may be positively affected by a lower amount of protein consumed by the infant [16]. It has been shown that excessive protein intake is a potential risk factor for the acceleration of renal function deterioration in children with mild chronic renal insufficiency [17]. The European Childhood Obesity Project (CHOP) study found that kidney size was increased in infants aged 6 months who were fed a higher-protein infant formula compared to lower-protein formula or breastfeeding [18]. As infants’ kidneys are still developing, given the potential adverse effects of a high protein intake on renal function deterioration, the European Food Safety Authority (EFSA) states that a low-protein intake during the first months of life is desirable. In addition, a high-protein intake early in life leads to increased urea production, which can impair the water balance of infants [19]. Given that lowering the protein intake reduces urea production [10], less protein intake via infant formula could also be beneficial in relation to this concern.

It needs to be noted that our study only addressed the effects after two years, so the effects of this nutritional intervention may appear in later childhood or even in early adolescence. Therefore, long-term follow-up of children participating in clinical trials assessing the effects of a low-protein infant formula is needed [20]. A possible explanation of the lack of effect on body composition could be that only very high protein intake increases the risk of obesity [21,22,23,24] (Figure 2). Hence, a threshold of early protein intake and its adverse effect on obesity risk might be possible. This threshold of protein concentration present in infant formula could be between 2.7 (higher-protein infant formula used in the Early Protein and Obesity in Childhood (EPOCH) study; no effect on obesity risk) and 4.4 g/100 kcal (higher-protein infant formula used in the CHOP study; negative effect on obesity risk), based on the available studies (Figure 2).

In addition to the physiological benefits, and from a completely different angle, a protein reduction in infant formula may positively affect the carbon footprint (CFP), a relative measurement of the amount of CO_2_ released in the environment during the life cycle of a product or activity, since food production, especially of dairy products, affects the environment in many ways, and dairy, meat, and eggs account for 83% of the greenhouse gas (GHG) emissions from the average EU diet [25,26]. Furthermore, since milk prices are based on the value of solids in the milk, with protein as one of the most expensive components of cow’s milk [26], it is plausible that a decrease in protein needs for the production of infant formula reduces both the cost price and the consumer price for infant formula.

**Figure 2 nutrients-14-02728-f002:**
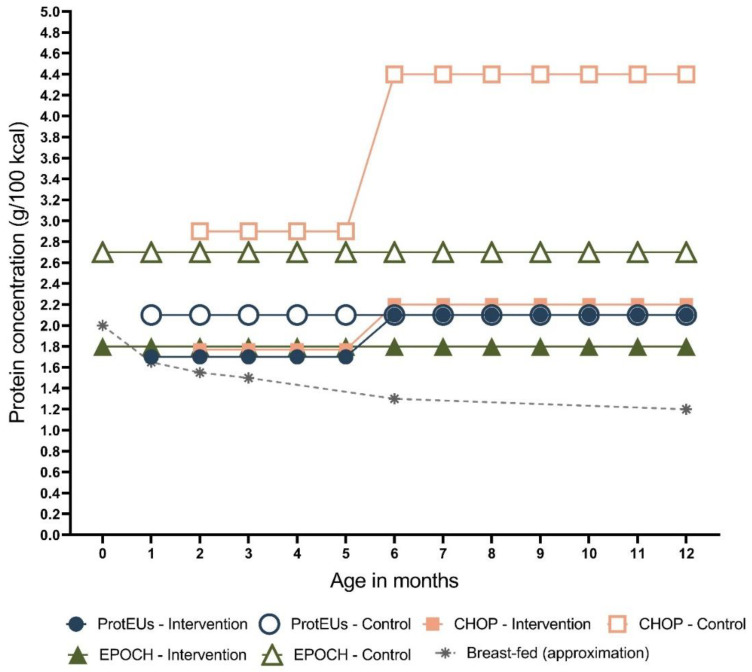
The protein concentrations of formulas used in CHOP [23], EPOCH [27], and ProtEUs [10] during the first 12 months of life (from 6 to 12 months, formula-fed infants in the ProtEUs were fed with standard follow-on formula). The protein concentrations in breast milk were adapted from a systematic review and meta-analysis of the nutrient content of term breast milk [28].

### Growth in Breastfed and Formula-Fed Infants

Even by lowering the protein levels, differences in growth, weight gain, and body composition are still found when comparing formula-fed infants to breastfed infants. In the ProtEUs study, the total fat mass, total fat-free mass, and fat-free mass index (FFMI) were significantly higher in both formula groups versus breastfed infants until the age of 6 months [10]. At the age of 2, significant differences were found in body composition between the children fed with low-protein infant formula and the breastfed reference group. In contrast, feeding children a standard infant formula did not result in body composition differences compared to breastfed children [11]. Contrary to these findings, the high-protein group included in the CHOP study had a significantly higher body fat percentage, fat mass index (FMI), and FFMI than the low-protein group at the age of 2 years. Furthermore, the breastfed infants in the CHOP study had the highest body fat percentage compared to both formula groups at that age [29]. The fat mass and fat-free mass (% of body weight) were similar in the formula and breastfed groups included in the EPOCH study at the ages of 1 and 3 years [27].

The mechanisms through which increased protein intake may affect growth and body composition remain to be clarified. As mentioned before, the anabolic hormones insulin and IGF-1 are responsive to fluctuations in protein intake [13] and have growth-stimulating properties [14,15]. Although no short-term associations among insulin, glucose, the Homeostatic Model Assessment for Insulin Resistance (HOMA-IR), and body composition have been described during the first year of life; these associations were seen at 2 years of age [12]. This may suggest that the reduction in insulin sensitivity early in life affects body composition at a later age rather than the opposite.

There is conflicting evidence comparing IGF-1 levels in formula-fed infants with different protein intake levels [12,27,30], and the association between IGF-1 and (the quality of) growth. The different findings highlight the complexity of the effect of IGF-1 on growth and body composition and the role of protein intake on IGF-1: both protein quantity and quality may be relevant to consider in this respect. The binding proteins (IGF-BP1, BP2, and BP3) and insulin sensitivity may play an important role in growth and body composition trajectories as formula-fed infants had lower insulin sensitivity compared to breastfed infants [12].

The higher growth rates and differences in the body composition of infants fed with low-protein infant formula compared to breastfed infants call for the use of infant formulas with an even lower protein content than 1.7 g protein per 100 kcal. Recently, it was shown that an infant formula with a protein amount of only 1.43 g/100 kcal resulted in a significantly lower weight gain rate than that of formula containing 1.9 g or 2.18 g protein/100 kcal during the first 4 months of life [31]. Although the investigators reported adequate growth, no broad conclusions can be reached regarding the safety of this very-low-protein infant formula. Importantly, the study formulas were not isoenergetic, and no data are yet available on the subjects’ body composition, volume intake, and blood parameters. Thus, before we can lower the protein intake to values closer to those found in human milk, more studies are needed on safety and efficacy.

## 3. Methodological Considerations Investigating the Effects of a New Infant Formula

### 3.1. The Study Formula

In general, it is crucial that both the investigational and the control formula are similar in every respect except for the component of interest. However, numerous reports show that infants can compensate for reduced (macronutrient) intake by increasing volume intake [27,32,33]. This may be assessed with food questionnaires, but the low accuracy and precision of food questionnaires are significant limitations. Unintentional inaccurate measurements and the over-dispensing of infant formula powder by the caregivers [34] are potential additional limitations. Therefore, additional techniques or methods are needed to measure the intake of infant formula (fluid intake) more precisely, for instance, by using stable isotopes [35]. However, this technique can be used for the measurement of fluid only. Therefore, this method cannot be used when, e.g., in addition to infant formula, complementary feeding is introduced.

Within studies investigating the effect of protein quantity or quality of infant formulas, the introduction of any other ingredients or additional modifications to experimental formulas, other than quantitative protein, could affect the obtained results [33,36]. As a consequence, findings cannot be attributed solely to differences in protein intake in such circumstances.

Using a control product with a protein level and quality that is based on the standard infant formula currently on the market makes it possible to compare the outcome of the intervention group with the result of a control group that reflects the current situation. This makes it easier to evaluate the clinical relevance of the study and its impact. Still, adequate potency is required in studies investigating one component of an experimental formula. On the other hand, designing an experimental study, developing a study formula, and conducting a clinical study requires at least 6 years before these results are available to the public. Adding several ingredients with a supposed beneficial effect on health outcomes may speed up improvements to infant formula, despite that one does not exactly know which ingredient is responsible for what effect.

### 3.2. Different Study Designs in Large Studies Addressing Protein Intake in Early Life

Different conclusions have been drawn after investigating the effect of protein intake early in life and its effect on later obesity risk. However, no effects of early protein intake on obesity risk in adults are available yet. The CHOP study found an effect of higher protein intake provided by infant formula during the first 12 months (Figure 2) of life on growth, body composition, and obesity risk until the age of 6. The EPOCH study, with a similar intervention period as the CHOP study, did not find an effect of different protein intake on growth and body composition until the age of 5. As argued before, this could be due to the amount of protein intake during early life (Figure 2).

At age 2, the CHOP, EPOCH, and ProtEUs studies describe different effects of protein intake on growth and body composition [10,11,22,24,27]. Whereas CHOP and EPOCH used formulas that did not differ substantially in quality, the ProtEUs study used a different approach. In the ProtEUs study, the amino acid content was based upon individually determined amino acid requirements, which led to a completely different amino acid profile. Despite this change, no effect on body composition nor BMI was observed. However, the ProtEUs study demonstrated that an infant formula with a modified amino acid composition and much lower protein intake during infancy generated adequate weight gain rates.

The start of the intervention might also be of importance. In general, the effects of nutritional interventions, if present, are likely to be larger the sooner the infant is exposed to the intervention. However, formula feeding may interfere with the optimal feeding choice: breastfeeding. The EPOCH study recruited infants within a week after birth (Figure 2, EPOCH [27]). Infants included in the CHOP study and the ProtEUs study were enrolled during the first 8 and 6 weeks of life, respectively, to ensure the parents were fully supported to breastfeed their infant (instead of starting with infant formula in an early phase). Cooperation with obstetrics departments provides the opportunity to inform parents about the possibility of participating in nutritional research as soon as possible after birth. However, home births are common in the Netherlands (14% of births in 2019 [37]) compared to other European countries such as Germany, France, Britain, and Belgium. Therefore, in addition, informing (future) parents in the prenatal phase would result in an earlier inclusion of infants into a nutritional study.

### 3.3. Protein Quality

Originally, protein formulas are derived from cow’s milk with an adapted whey-to-casein ratio. Despite this adaptation, amino acid concentrations differ substantially from that of breast milk.

For healthy term-born infants, up until the age of 6 months, the assumption is that human milk from a well-nourished mother can be regarded as providing optimal intake for the infant. These data can guide the composition of infant formula. However, compositional similarity is not an adequate determinant nor indicator of the safety and differences in nutritional adequacy/bioavailability and digestibility, and the matrix complexity must be taken into account. Therefore, the nutritional requirements of formula-fed infants should be defined on the basis of experimental studies. For amino acids or protein, the nitrogen balance method is used to investigate the difference between nitrogen intake and the amount excreted in urine, feces, skin, and miscellaneous losses, such as breath and sweat. Since it is considered unacceptable to maintain infants and children on either deficient or excessive intake, alternative methods were needed. The availability of isotopically labeled tracers made it possible to determine the metabolic fate of a labeled amino acid at varying dietary intake levels. An approach to estimating indispensable amino acid requirements using amino acid oxidation is the indirect amino acid oxidation (IAAO) technique. The advantages of the IAAO method are the short adaptation time of the study diet (1–3 days), and no dietary restrictions on the intake of the test amino as a result of the fact that the requirement of one amino acid is determined by the oxidation of another amino acid. This makes it possible to study all possible dietary levels of essential amino acids. Within the ProtEUs study, a customized blend of essential amino acids was used in the investigational group. The amino acid composition present in this infant formula was based on the outcomes of clinical trials conducted in healthy term formula-fed infants using the IAAO technique. However, we showed that adapting both the protein quantity and quality of infant formula did not affect infants’ growth or body composition.

### 3.4. Growth Adequacy Evaluation and Safety

Mostly, modifications or the addition of an ingredient new to infant formulas are driven by the manufacturer’s desire to produce a formula that mimics the advantages of breastfeeding when provided to the child. The safety of new infant formulas will need to be judged against two controls: the previous version of the formula without the added ingredient and human milk.

Overall, in clinical trials, infant growth is used as a safety parameter. The growth of infants fed lower-protein infant formula is compared with that of standard infant formula and breastfed controls. It is difficult to decide whether the growth of breastfed or standard formula-fed infants is the reference growth in these studies.

The World Health Organization (WHO) describes the growth of carefully selected and exclusively breastfed infants as the reference growth for infants until the age of 6 months. However, assessing the safety of lower-protein infant formula by comparing the growth of infants fed this proposed formula to the growth of breastfed infants has both regulatory and research issues. Clinical studies that assess the effects of infant formulas with new ingredients or compositions are difficult to design as infants cannot be randomized to consume formula or human milk. It is therefore difficult to sort out which of the factors of breastfeeding are due to nutritional components and which are accounted for by social and psychological factors. Obviously, randomized trials assigning infants to breastfeed or formula-feed are not ethically feasible.

The American Academy of Pediatrics (AAP) concluded that the weight gain rate is the single most valuable component of the clinical evaluation of infant formula [38] and recommended that growth studies conducted for the purpose of assessing new infant formulas should be able to detect a difference of 3.0 g of weight gain per day between the investigational infant formula group and the control group (standard infant formula) during the first 3 to 17 weeks of life [38,39]. However, the higher weight gain rate in formula-fed infants during the first months of life compared to breastfed infants might be the cause of higher weight-for-length, BMI, and obesity risk in later life. Therefore, it might be desirable to lower the weight gain rate of formula-fed infants.

The evaluation of infants’ growth using anthropometric data converted to *z* scores using the WHO Child Growth Standards [40] makes it possible to assess whether infants’ growth is within a normal range. It must be taken into account that these growth curves are based on healthy breastfed infants with different growth characteristics because of their widely diverse cultural backgrounds. Furthermore, the infants were exclusively or predominantly breastfed for at least 4 months only.

## 4. Future Perspectives and Implications—Infant Feeding

### 4.1. Improving Infant Formulas

Infant formulas have been designed to mimic human milk for infants who cannot be breastfed. The overall goal is to establish similar functional outcomes to assure optimal growth, development, maturation of the immune system, and programming of the metabolic system. The potential advantages of breastfeeding are the reasons behind many of the proposed addition of ingredients to infant formulas. However, after decades of improving infant formula, growth pattern [41] and body composition development [42] are still different in formula-fed infants compared to breastfed infants, which could contribute to an increased risk of obesity among formula-fed infants.

In 2017, the EFSA concluded that follow-on formula with a protein content of at least 1.6 g/100 kcal from intact cow’s or goat’s milk protein, otherwise complying with the requirements of relevant EU legislation, is safe and suitable for healthy infants living in Europe [43]. Our findings call for the use of infant formulas with an even lower protein content than 1.7 g protein per 100 kcal. In line with follow-on formula, it is likely that a protein content of at least 1.8 g/100 kcal of infant formula will also be lowered in the upcoming years.

Based on the protein concentrations present in breast milk (Figure 1) and to better approach the protein needs of formula-fed infants, it has been proposed that different formulas could be created for the first half year of life with different protein concentrations (Figure 3) [44]. This staging concept could achieve growth and metabolic outcomes more similar to that of breastfed infants. However, additional studies on amino acid requirements in formula-fed infants need to be performed at different ages.

From the introduction of the first infant formula onwards, milk protein and its processes have been optimized to provide the most optimal amino acid profiles in infant formula. However, based on requirement studies [45,46,47,48,49,50], there is still no intact protein to provide the right composition of essential amino acids. Therefore, the addition of free amino acids is unavoidable, which has its disadvantages. The use of alternative proteins for infant formula, such as egg protein or plant protein, could be considered but need adequate testing for safety and efficacy.

### 4.2. Plant Protein-Based Infant Formula

Many studies have shown the big impact of the food industry on the environment. Animal-based food production generally has a bigger climate impact than plant-based foods due to higher emissions from its production, manure management, and enteric fermentation [51,52,53].

Although the options for replacing animal protein with plant protein in an adult diet are abundant, in infant formula, these options are very limited. An important reason for this is the high quality of protein present in bovine milk, along with high digestibility [54]. In contrast, plant proteins are of low quality and frequently lack one or more essential amino acids. Moreover, they usually have low digestibility. An example of a promising plant protein for infant formula that warrants further investigation is isolated quinoa proteins (IQPs) as the protein quality is relatively high. Moreover, during its protein isolation process, unfavorable compounds are mostly removed, ensuring that the final product can comply with the maximum residue concentrations allowed. More research is needed before the introduction of IQP in infant formulas is considered, but it has several promising features that warrant further investigation [55].

The replacement of animal proteins with plant proteins involves a lot of critical aspects in order to optimize the effect of processing on product quality. For instance, the heating step needs improvement. In addition, the protein isolates obtained from different plants have a different color than the conventional protein sources used in infant formula. Moreover, the taste of plant products differs a lot from the currently used milk sources. Both parents and infants need to accept these changes and get used to the taste and appearance of the formula.

A practical aspect that needs to be taken into account is the certainty of plant product availability. A stable, long-lasting supply of both the agricultural company and the protein manufacturer must be secured.

### 4.3. Promising Interventions and Preventive Strategies

It has been suggested that subgroups could benefit from a lower-protein intake during the first months of their lives. An example of such a subgroup is infants born to overweight or obese mothers since they are at increased risk of being overweight later in life [56,57,58,59,60,61,62]. A reason for this could be the accelerated growth during the first year of life of infants born to obese mothers [63] since rapid weight gain in infancy is associated with later overweight and obesity [64,65]. Some years ago, it was already shown that an infant formula with a lower protein content slows weight gain in infants aged 3 to 12 months of overweight and obese mothers [66].

To investigate if low-protein infant formula has the potential to reduce the risk of future obesity in this group of infants, we are establishing a large multicenter long-term randomized controlled trial in infants at risk of adiposity, e.g., infants born to overweight or obese mothers. Within this trial, we will investigate the effects of a new infant formula with a protein level closer to breastmilk on growth and body composition. Moreover, the protein used in this investigational infant formula will be partly derived from plants, and the formula will be composed of a high level of intact protein (only 10–20% will be derived from free amino acids). As mentioned before, the transition from animal-based protein to plant-based protein aims to have a positive outcome on the environment, which has been mentioned before. By feeding infants this formula, we not only expect to improve growth and body composition development in formula-fed infants but also to positively influence the climate impact of the food industry.

It seems that preventive strategies for reducing childhood obesity should focus on maternal BMI rather than on pregnancy complications such as gestational diabetes and gestational hypertensive disorders [67]. Therefore, behavioral interventions in an overweight or obese pregnant woman might improve clinical outcomes in the child. However, there is conflicting evidence as to whether diet and physical activity in these women would be effective and reduce the incidence of large-for-gestational-age infants, thereby affecting the risk of childhood obesity [68,69,70,71]. Large, high-quality studies including long-term follow-up are warranted to answer these research questions.

The adverse effects of a maternal diet with a high glycemic index increase the glucose transfer to the fetus, and, consequently, fetal growth and adiposity [72,73], particularly among overweight or obese pregnant women who are more likely to consume diets rich in carbohydrates and to have impaired glucose tolerance [74]. Recently it has been shown that among overweight and obese women and their children, a higher maternal early-pregnancy dietary glycemic index was associated with increases in childhood BMI and fat accumulation [75]. Therefore, targeting the dietary glycemic index among overweight and obese pregnant women may be a promising tool to help prevent childhood adiposity in the future.

### 4.4. The Importance of Breastfeeding Support

We can adjust and further improve infant formula, but we will not be able to mimic breast milk or the art of breastfeeding because of its complexity and its effect on infant physiology, including reducing the risk of obesity, among other benefits [76]. Breast milk is the gold standard of infant nutrition and provides a mixture of unique components required for the optimal development of the infant. Breast milk is beneficial for short- and long-term health of both infants and their mothers. Therefore, the support and promotion of breastfeeding by healthcare professionals are of paramount importance.

Based on the current evidence, it is necessary to execute new trials evaluating infant formulas with improved protein quality together with further reductions in protein content. In addition, further research should study possible underlying mechanisms of early protein intake on later health and the most optimal protein source for infant formula for optimal growth and body composition.

## Figures and Tables

**Figure 1 nutrients-14-02728-f001:**
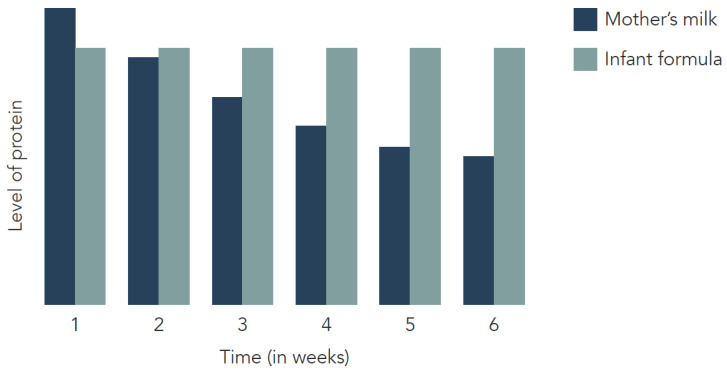
Protein concentrations of infant formula and breast milk over time (schematically).

**Figure 3 nutrients-14-02728-f003:**
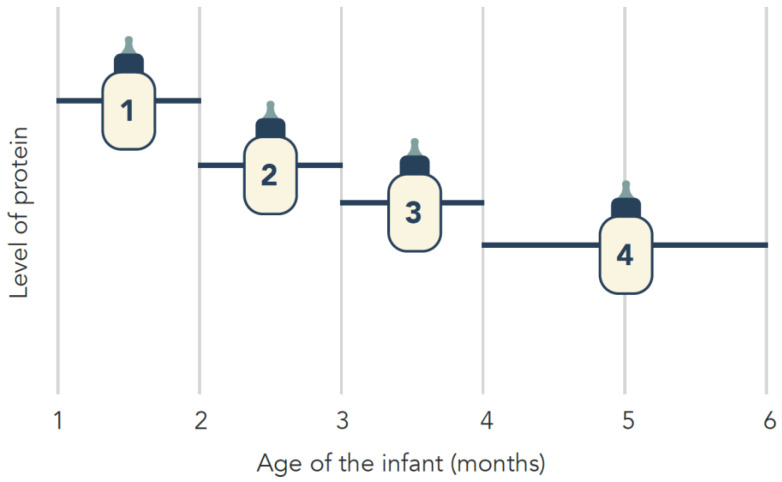
Staging concept: infant formula during the first 6 months of life.

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
