# Peer review of "Low-Protein Infant Formula and Obesity Risk"

_nutrients, 2022, doi:10.3390/nu14132728_

Round 1

Reviewer 1 Report

The article describes an important topic on clinical nutrition: low-protein infant formula and obesity risk. The authors say ” Therefore a lower protein intake could possibly prevent childhood obesity”, which to my understanding is a hypothesis rather than a conclusion. However, without reading the text, readers can easily mistake this sentence as the conclusion. So I strongly suggest the authors to clearly state their conclusion in their abstract, even the conclusion is “inconclusive”.

Although there seems to be a lack of published work on the topic for a more meaningful review.

Minor:

Line 30: risk of morbidity…

Line 374: gold standard

Reviewer 2 Report

Infant formulas are a well-known substitute for human milk, and it is used for the nutrition of infants who cannot be breastfed. The goal of this paper was to present the current data in the field about how the protein concentration of protein in infant formulas can relate to the obesity risk of infants. In their conclusion, they suggest new studies that could improve the understanding of the effect of low protein infant formula on the health of infants.

General comments:

The work presented in the paper is relevant to the field and will benefit the scientific field of pediatric nutrition.

In section 2 (“Could lower protein intake be a mediator in the reduction of obesity?”), the authors briefly talk about how reducing the protein concentration of infant formula may affect carbon footprint by decreasing the dairy products used to produce it (Line 89-96). This paragraph seems out of place because this topic is mentioned later in the paper section 4. I would suggest the authors create a new section to discuss it in more detail to strengthen the later discussion in section 4.

In addition, in section 3 the authors mentioned that randomized trials would not be ethically feasible. Can retrospective studies use existing patient data to answer some of these questions?

Specific comments:

The first time using an abbreviation it should write the full name followed by the abbreviation: ProtEU (Line 56); IGF (Line 65); CHOP (Line 80); EPOCH (Line 105); FFMI (Line 111); FMI (Line 117); HOMA-IR (Line 127); WHO (Line250); RCT (Line343).

Figure legends: it should add the data source. Since it is a review paper all data was collected from the published literature that should cite in the legend of the figures.
